# The Ubiquitin Proteasome System Is a Key Regulator of Pluripotent Stem Cell Survival and Motor Neuron Differentiation

**DOI:** 10.3390/cells8060581

**Published:** 2019-06-13

**Authors:** Monique Bax, Jessie McKenna, Dzung Do-Ha, Claire H. Stevens, Sarah Higginbottom, Rachelle Balez, Mauricio e Castro Cabral-da-Silva, Natalie E. Farrawell, Martin Engel, Philip Poronnik, Justin J. Yerbury, Darren N. Saunders, Lezanne Ooi

**Affiliations:** 1Illawarra Health and Medical Research Institute, Northfields Avenue, Wollongong, NSW 2522, Australia; mcb675@uowmail.edu.au (M.B.); pddh859@uowmail.edu.au (D.D.-H.); stevensc@uow.edu.au (C.H.S.); sh659@uowmail.edu.au (S.H.); rb478@uowmail.edu.au (R.B.); mcastro@uow.edu.au (M.eC.C.-d.-S.); nfarrawe@uow.edu.au (N.E.F.); mengel@uow.edu.au (M.E.); jyerbury@uow.edu.au (J.J.Y.); 2School of Chemistry and Molecular Bioscience, University of Wollongong, Northfields Avenue, Wollongong, NSW 2522, Australia; 3School of Medical Sciences, University of New South Wales, Sydney, NSW 2052, Australia; jess.mckenna91@gmail.com; 4School of Medical Sciences, Faculty of Medicine and Health, The University of Sydney, Camperdown, NSW 2050, Australia; philip.poronnik@sydney.edu.au

**Keywords:** ubiquitin, ubiquitinomics, UBA1, induced pluripotent stem cell, motor neuron, motor neurone disease, amyotrophic lateral sclerosis

## Abstract

The ubiquitin proteasome system (UPS) plays an important role in regulating numerous cellular processes, and a dysfunctional UPS is thought to contribute to motor neuron disease. Consequently, we sought to map the changing ubiquitome in human iPSCs during their pluripotent stage and following differentiation to motor neurons. Ubiquitinomics analysis identified that spliceosomal and ribosomal proteins were more ubiquitylated in pluripotent stem cells, whilst proteins involved in fatty acid metabolism and the cytoskeleton were specifically ubiquitylated in the motor neurons. The UPS regulator, ubiquitin-like modifier activating enzyme 1 (UBA1), was increased 36-fold in the ubiquitome of motor neurons compared to pluripotent stem cells. Thus, we further investigated the functional consequences of inhibiting the UPS and UBA1 on motor neurons. The proteasome inhibitor MG132, or the UBA1-specific inhibitor PYR41, significantly decreased the viability of motor neurons. Consistent with a role of the UPS in maintaining the cytoskeleton and regulating motor neuron differentiation, UBA1 inhibition also reduced neurite length. Pluripotent stem cells were extremely sensitive to MG132, showing toxicity at nanomolar concentrations. The motor neurons were more resilient to MG132 than pluripotent stem cells but demonstrated higher sensitivity than fibroblasts. Together, this data highlights the important regulatory role of the UPS in pluripotent stem cell survival and motor neuron differentiation.

## 1. Introduction

Ubiquitin is a small 8.6 kDa protein that can be covalently linked, either as a monomer or as a polyubiquitin chain, to protein lysine residues. Ubiquitylation is one of the most abundant protein modifications in cellular signaling, controlling numerous cellular pathways, and is a key regulator of protein homeostasis (proteostasis) [1]. Depending on the length and structure of the ubiquitin chain, protein ubiquitylation either tags substrate proteins for degradation via the proteasome (the ubiquitin-proteasome system; UPS) or alters substrate protein localization or function [2]. The 76 amino acid long ubiquitin (Ub) protein is conjugated to substrate proteins via an ATP-dependent hierarchical cascade involving an E1 ubiquitin-activating enzyme, E2 conjugating enzymes, and E3 ubiquitin ligases [3,4]. A key characteristic of Ub is its ability to form poly-Ub chains through sequential conjugation of Ub molecules via lysine residues, with diversity in Ub chain topology controlling the fate of substrate proteins, analogous to a cellular code [4]. For example, poly-Ub chains linked via lysine residues K48 or K11 of the ubiquitin molecule signals delivery of substrate proteins to the proteasome system for degradation. Up to 90% of cellular proteins are degraded through this system [5].

Dysfunction of the UPS is involved in diverse diseases, including neurodegenerative diseases, cystic fibrosis and cancer [6]. Embryonic stem cells are derived from the inner cell mass of an embryo during early development. As such, they are able to give rise (differentiate) to almost all cell types in the body, a property described as pluripotency. Induced pluripotent stem cells (iPSCs) are cells that have been reprogrammed from somatic cells, such as fibroblasts, into a pluripotent state [7]. Because iPSCs can be derived from individuals with various diseases or from healthy individuals, iPSCs have attained extensive application in disease modelling, developmental studies and cellular replacement therapies [8]. Despite having a key role in many aspects of fundamental cell biology, the role of protein ubiquitylation in pluripotency and differentiation has not been widely characterised. Understanding the cellular processes that govern motor neuron differentiation from iPSCs is crucial to further our understanding of differentiation, as well as motor neuron diseases. During the past decade, there have been vast improvements in motor neuron differentiation protocols, leading to relatively high yields of pure motor neurons in culture. Optimization of protocols has increased efficiency of motor neuron generation to 90% [9] from initial attempts producing just 20% [10]. These neurons express motor neuron specific markers, including HB9, Islet1 and SMI32 [11,12]. High purity cultures have significantly improved our ability to examine the dynamic signaling, metabolic and functional landscape of motor neurons. A greater understanding of motor neuron differentiation is paramount to further development of differentiation protocols, and will be critical to the potential application of these cells in tissue replacement therapies. Motor neurons derived from iPSCs will therefore likely play an important role in identifying underlying mechanisms in motor neuron diseases.

Increasing evidence suggests the ubiquitin proteasome system (UPS) is dysregulated in a range of neurodegenerative diseases and UPS dysfunction underlies motor neuron susceptibility in amyotrophic lateral sclerosis (ALS) [1]. Mutations in UPS-associated genes have been shown to cause ALS, including the E3 ubiquitin ligase *CCNF* [13], *UBQLN2* [14], *VCP* [15] and *SQSTM* [16]. Altered Ub distribution has been observed in ALS models [17], and mutations in the key UPS regulator ubiquitin-like modifier activating enzyme 1 (UBA1) cause the juvenile motor neuron disease, spinal muscular atrophy (SMA) [18,19]. Conversely, increased expression of UBA1 has been shown to attenuate the disease phenotype of SMA in mouse and zebrafish models [20,21]. Spinal motor neurons appear more susceptible than other cell types to UPS stress [22] and this may be due to a reduced ability to upregulate the UPS in response to stress.

Quantitative proteomics provide an unbiased approach that can detect protein changes during disease or development. In stem cell pluripotency and differentiation, proteomic analyses have identified systems-level mechanisms controlling cell fate and the coordination of events that are necessary to drive specific development [23]. Protein–protein interactions and post-translational modifications can be identified by combining affinity purification with proteomic identification of interacting proteins. These approaches have increased our understanding of the protein signaling networks involved in pluripotency and cellular differentiation [24]. Examination of the ubiquitin-modified proteome (ubiquitome) using mass spectrometry provides an unbiased and systematic approach to characterize changes in the spectrum of ubiquitylated proteins present in different cellular stages and differentiation states. While Buckley et al. [25] previously investigated the ubiquitome in mouse iPSCs in comparison with a combination of mouse embryonic stem cells and mouse embryonic fibroblasts, to the best of our knowledge, no studies have yet studied the ubiquitome of human iPSC-derived motor neurons compared to their derivative cells in a feeder-free system. Hence, we employed quantitative proteomics (ubiquitomics) to map the ubiquitome of human iPSCs and iPSC-derived motor neurons. Ubiquitomics showed dynamic changes in the ubiquitome across these cellular stages, supporting a key role for Ub signaling in both pluripotency and differentiation. Further, we identified a remarkable susceptibility of iPSCs and iPSC-derived motor neurons to UPS inhibition, compared to fibroblasts from the same donors. Together our data identifies the network of proteins regulated by ubiquitin and highlights the essential role of the UPS in stem cell survival and motor neuron differentiation.

## 2. Materials and Methods

### 2.1. Generation of iPSC-Derived Motor Neurons

All cells were maintained in a humidified 5% CO_2_ (*v*/*v*) incubator at 37 °C. Fibroblasts were acquired from healthy human donors via skin biopsy. Fibroblasts were routinely cultured in DMEM/F12 supplemented with 10% (*v*/*v*) fetal bovine serum (Interpath, Heidelberg West, VIC, Australia) and 10% PenStrep (*v*/*v*; Life Technologies, Carlsbad, CA, USA) with three media changes per week.

The induced pluripotent stem cell lines used in this study were generated from a healthy 57 year old male [26], a healthy 56 year old female [27], and a healthy 59 year old female (details described below). H9 embryonic stem cells were also used in this study and cultured using the same protocol as the iPSCs. Fibroblasts were reprogrammed into induced pluripotent stem cells using microRNA enhanced mRNA reprogramming as per the manufacturer’s recommendations (2nd Generation reprogramming kit mRNA Reprogramming Factor Set #00-0067 and microRNA Reprogramming Cocktail #05-0036, Stemgent, Cambridge, MA, USA) and stem cells were characterised as pluripotent as we have previously carried out [27,28]. Following expansion of several viable clones from the reprogrammed cells, iPSCs were maintained on Matrigel-coated tissue culture plates in TeSR E8 (Stem Cell Technologies, Tullamarine, VIC, Australia) and passaged upon nearing 80% confluency, using dispase (1 mg/mL; Stem Cell Technologies) to be further characterised. Clones were selected and further characterised for pluripotency by PluriTest and immunocytochemistry to confirm the expression of pluripotency markers [29]. The iPSC colonies were cultured on Matrigel (Sigma Aldrich, St. Louis, MO, USA) coated coverslips for one day following passage. Immunocytochemistry was performed as previously described [27,28]. Briefly, the antibody details were: OCT4 (1:1000; mouse; 01550, AB_1118539 Stem Cell Technologies), TRA-1-60 (1:200; rabbit; 130-100-832, Miltenyi Biotec Macquarie Park, NSW, Australia), Alexa Fluor 488 Goat anti-mouse IgG (1:1000; A11001 AB_2534069, Thermo Fisher, North Ryde, NSW, Australia) and Alexa Fluor 488 Goat anti-rabbit IgG (1:1000; A11008, AB_143165, Thermo Fisher). Images were captured on an epifluorescence microscope (Leica DMI8 or DMI6000B) acquired using LAS AF (Leica Microsystems, Macquarie Park, NSW, Australia). The iPSC karyotyping was performed for each reprogrammed line through StemCore (University of Queensland, Brisbane, Queensland, Australia) to confirm the absence of introduced chromosomal abnormalities.

### 2.2. Motor Neuron Differentiation and Characterisation

Human donor-derived iPSCs were differentiated to motor neurons as described in [30,31]. Neural precursors were plated onto collagen I (10 μg/mL; Life Technologies), laminin (20 μg/mL; Invitrogen, Carlsbad, California, USA) and fibronectin (10 μg/mL; Life Technologies) coated tissue culture plates. Precursors were caudalized with retinoic acid (0.3 µM; Sigma Aldrich) for three days, followed by one week of culture with retinoic acid (0.1 µM; Sigma Aldrich) and purmorphamine (2 µM; Stem Cell Technologies). The motor neurons were matured for four weeks with partial media changes every other day. Motor neuron yield was determined through motor neuron specific marker expression via immunocytochemistry, using antibodies against neurofilament heavy (SMI32; 1:800; rabbit; ab73273, Abcam, Cambridge, UK), Islet 1 (1:200; mouse; AB4326, Merck Millipore, Burlington, MA, United States) and HB9 (1:50; mouse; Developmental Studies Hybridoma Bank, University of Iowa, Iowa City, IA, USA).

### 2.3. Sample Collection

Cells were harvested by first gently washing in PBS, followed by manually scraping in lysate buffer with deubiquitylating inhibitor (50 mM Tris-HCl (pH7.4; Sigma Aldrich), 1% sodium deoxycholate (Sigma Aldrich), 150 mM NaCl (Sigma Aldrich), 1 mM EDTA (Sigma Aldrich), 1% TritonX-100 (Sigma Aldrich), 0.1% SDS with 1 mM sodium orthovanadate (Sigma Aldrich), 10 mM N-Ethylmaleimide (Sigma Aldrich), 1× Phosphate inhibitor cocktail 2 (Roche, Basel, Switzerland).

### 2.4. Ubiquitomics

Ubiquitin proteomic analysis was performed on cell lysates as previously described [32]. Protein concentration was determined using a BCA assay and 500 μg of total protein was used per sample to immunopurify mono- and poly-ubiquitinated proteins using specialized ubiquitin affinity matrix (VIVAbind Ubiquitin Kit, VIVA Bioscience, Exeter, UK). After substantial washing to remove residual detergent, beads were digested for 30 min at 27 °C, then reduced with 1 mM DTT and left to digest overnight at room temperature with sequencing-grade trypsin (5 μg/mL, Promega, Madison, WI, USA). Samples were alkylated with 5mg/mL iodoacetamide and protease digestion terminated with trifluoroacetic acid. Trypsinized eluents were collected after brief centrifugation then purified and desalted using self-packed tips with 6 layers of C18 Empore disks (Pacific Laboratory Products), then dried in a SpeedVac. Samples were then resuspended in 12 μL 5% formic acid, 2% acetonitrile and stored at −80 °C. Five microlitres of each digested peptide sample was loaded and fractionated along a custom C_18_ column and introduced by nanoelectrospray into an LTQ Orbitrap Velos Pro coupled to an Easy-nLC HPLC (Thermo Fisher). Tandem mass spectrometry data was collected for the ten most abundant ions per scan over a 140 min time gradient and fragmented within the linear ion trap using higher-energy collisional dissociation. The order of data collection was randomized to interchange between biological conditions with BSA run between each sample to minimize temporal bias. Ms/MS data were searched using MaxQuant (v1.2.7.4, Max Planck Institute of Biochemistry, Martinsried, Germany [33]) against the Uniprot human database (release 2016_04). A false discovery rate of 1% was tolerated for protein, peptide, and sites, and one missed cleavage was allowed. Sonifications of proteomic data distributions were compared [34]. Data were filtered for contaminants, reverse hits, proteins only identified by site and number of unique peptides (≥ 2). Statistical and functional analysis was performed as previously described [30] using InteractiVenn (Sao Paolo, Brazil) [35], STRING (v10.5) (Zurich, Switzerland) [36], Cytoscape (v3.6.1) (San Diego, CA, USA) [37,38], and PRISM (v7, Graphpad, San Diego, CA, USA). For nonquantitiave analysis, proteins were defined as “present” in either condition if detected in a least two replicates.

### 2.5. Ubiquitin Proteasome System Inhibition

To examine the effects of UPS stress, cells were treated with the proteasome inhibitor MG132 (concentration curve 0 (DMSO vehicle control), 1, 2.5, 10, 100 and 250 μM; Sigma Aldrich). In the vehicle control, DMSO was used at the same final concentration in media (0.1%) as in the MG132 treatments. PrestoBlue, a resazurin based assay, was used as a measure of cellular viability, as per the manufacturer’s instructions (Thermo Fisher). Cells were incubated in MG132 for 16 h before PrestoBlue reagent was incubated with the cells for 2 h. The PrestoBlue supernatant was measured for fluorescence at 560 nm excitation/590 nm emission. Images of iPSCs over the course of incubation with MG132 were taken using an Incucyte Zoom (Essen Bioscience, Ann Arbor, MI, USA). Inhibition of UBA1 in motor neuron precursors via PYR41 (Sigma Aldrich) was first investigated via incubation in media at a final concentration of 0 μM (DMSO vehicle control), 1 μM, 2.5 μM, 5 μM, and 10 μM for 24 h. In the vehicle control DMSO was used at the same final concentration in media (0.1%) as in the PYR41 treatments. Motor neuron precursors were treated with PYR41 with a partial media change and imaged using the Incucyte Zoom on alternate days over the normal course of differentiation. Images from the course of differentiation were analyzed for neurite outgrowth using the Incucyte Zoom NeuroTrack package. Cell-body cluster was set to segmentation mode (0.9 adjustment) and neurite filters were set to best, with 0.6 neurite sensitivity. A PrestoBlue viability assay was performed on the PYR41 treated motor neurons, as previously described.

### 2.6. Statistics

Data shown are mean ± SEM and were assessed by *t*-test, one-way or two-way ANOVA as appropriate with post hoc correction as stated in the text, *p* < 0.05 was considered significant.

## 3. Results

### 3.1. High Yield Differentiation of Motor Neurons from Fibroblast-Derived iPSCs

To investigate molecular mechanisms underlying differentiation of motor neurons in development, and as potential models for diseases involving motor neurons, fibroblasts were reprogrammed to iPSCs and differentiated into motor neurons (Figure 1A,B) [27]. Reprogrammed cells from human donor-derived fibroblasts displayed characteristic stem cell-like morphology (Figure 1B) and colonies were found to ubiquitously express protein markers of pluripotency, including OCT4 (Figure 1C) and TRA-1-60 (Figure 1E), by immunocytochemical analysis and showed normal karyotypes (Figure 1D). Cells tested positive as pluripotent in the PluriTest (Figure 1F) [29]. The iPSCs were differentiated towards a spinal motor neuron lineage. Differentiated cells exhibited motor neuron morphology, with a large soma, expression of the motor neuron-specific marker and long neurites that extended over 400 μm in length, as shown by neurofilament heavy expression, SMI32 (Figure 1G) and Islet1 (Figure 1H) and HB9 in the nucleus (Figure 1I). Differentiated cells expressed the motor neuron-specific nuclear markers HB9 and Islet1, detected by immunocytochemistry at yields of 88.1 ± 1.4%, and 90.5 ± 1.4%, respectively (Figure 1H,I).

### 3.2. The Ubiquitin-Modified Proteome (Ubiquitome) of iPSCs and Motor Neurons

To gain insight into the role of the ubiquitin-mediated regulation of iPSCs and iPSC-derived motor neuron biology, we sought to determine changes to the ubiquitin-modified proteome (ubiquitome) during differentiation of motor neurons from iPSCs. The iPSC and iPSC-derived motor neuron lyastes were subjected to affinity enrichment and quantitative liquid chromatography tandem mass spectrometry (LC-MS/MS) (Figure 2A) to identify ubiquitin-modified proteins. LFQ intensities were confirmed as normally distributed (Figure 2B,C). We observed a significantly expanded ubiquitome in motor neurons (1046 ubiquitin-modified proteins) compared with iPSCs (535 ubiquitin-modified proteins) (Figure 2D and Appendix A). We identified 439 ubiquitlyated proteins that were common to both cell states, although quantitative analysis revealed many of these were represented at different levels (see below). We observed strong correlations between individual protein label-free quantification (LFQ) intensities between iPSC replicates generated from multiple passages of the same iPSC line, and between motor neuron replicates generated through separate differentiations (Figure 2E). We also observed significant overlap in proteins identified in individual replicates (Figure 2G). Hierarchical clustering showed that ubiquitome profiles of individual iPSC or MN replicates were more similar to each other than to samples from the other condition (Figure 2E).

In motor neurons, there were significantly elevated levels of Ub-associated forms of a number of key enzymes at each level in the Ub conjugation hierarchy (E1, E2, E3 and DUB) (Figure 2F), suggesting a general activation of the Ub system in the differentiated motor neurons compared to pluripotent stem cells. To ensure consistency between samples, proteomics data is normalized to protein content at both the affinity purification and LC-MS/MS stages of analysis. This normalization is evident in the equal representation of various histones in the ubiquitomes of iPSCs and motor neurons, since ubiquitylated histones represent a large proportion of the total cellular ubiquitin pool. We observed no significant difference in the ubiquitylation of individual histones between iPSCs and motor neurons (Figure 2H) or the majority of individual proteasome subunits, with the exception of numerous “delta” subunits, which were not detected in the iPSC ubiquitome (Figure 1J). Importantly, we observed very little difference in the total amount of ubiquitin detected in both states (Figure 2F,H) and protein input was normalized at both the affinity purification and LC-MS/MS stages of analysis. Furthermore, the ubiquitomics approach deployed here was not able to distinguish between conjugated or “free” ubiquitin. Hence, even though the total cellular pool of ubiquitin is similar between iPSC and motor neuron states, there appears to be enhanced or activated Ub modifying (ubiquitylation) activity in motor neurons.

KEGG classifiers were used to analyses functional enrichment of proteins in the ubiquitome of iPSCs and motor neurons. Top scoring pathways in the ubiquitome of both iPSCs and motor neurons included ribosome (hsa03010), metabolic pathways (hsa01100), spliceosome (hsa03040), proteasome (hsa03050), and RNA transport (hsa003013) (Figure 2I, Appendix A). Pathways enriched specifically in the iPSC ubiquitome included focal adhesion (hsa04510) and cell cycle (hsa04110), while specifically enriched pathways in the motor neuron ubiquitome included endocytosis (hsa04144), ubiquitin mediated proteolysis (hsa04120), neurodegenerative conditions (Huntington’s disease (hsa05016), Parkinson’s disease (hsa05012) and Alzheimer’s disease (hsa05010)), cell–cell interaction (gap junction (hsa04540), tight junction (hsa04530), adherens junction (hsa04520)) and synaptic function (dopaminergic synapse (hsa04728) and synaptic vesicle cycle (hsa04721)) (Figure 2I, Appendix A). The broad representation of numerous proteins implicated in motor neuron function in the ubiquitome of these cells indicates an important role for Ub in regulating motor neuron differentiation and function. Consistent with this observation, we observed a significant enrichment of a number of components of the ubiquitin proteasome system in the motor neuron ubiquitome, including E1 ubiquitin activating enzyme UBA1, E2 conjugating enzymes (UBE2O, UBE2L3, UBE2N), E3 ligases (NEDD4L, RNF25, DDB1), DUBs (USP5, USP7, OTUB1) (Figure 2F), and delta subunits of the proteasome (PSMD1, 6, 7, 10, 12 and 13) (Figure 2J).

### 3.3. Quantitative Analysis of the Changing Ubiquitome During Motor Neuron Differentiation

Quantitative analysis revealed widespread changes to the abundance of ubiquitylated proteins (using a cut off of fold change > 2, FDR < 0.01) following differentiation of iPSCs to motor neurons, with 585 ubiquitylated proteins more abundant in motor neurons, and 120 ubiquitylated proteins decreasing in abundance during differentiation (Figure 3A and Appendix A). Using STRING to map known protein-protein interactions among differentially ubiquitylated proteins during differentiation of iPSCs to motor neurons highlighted significant enrichment of a number of functional pathways (Figure 3B,C). Proteins with increased abundance in the motor neuron ubiquitome are associated with numerous neuronal functions, including synaptic transmission (e.g., DNM2, KIF5B, KIF5C), and glucose metabolism (glycolysis (e.g., CS, FH, PDHA1), TCA cycle (e.g., PFKL, PGM1, ALDH9A1) and oxidative phosphorylation (e.g., NDUFS1, ATP5D). Individual protein abundances from selected enriched pathways are shown in Appendix A. Alterations in the abundance of individual proteins within enriched functional groups (KEGG pathways) in synaptic proteins, metabolism, cell/extracellular matrix interactions and the spliceosome are shown in Appendix A.

### 3.4. Differential Sensitivity of iPSCs and Motor Neurons to UPS Inhibition

Given the widespread involvement of the UPS in various cellular pathways in both iPSCs and motor neurons, we sought to determine the effects of UPS inhibition on survival of each of these cell types. First, we compared the viability of iPSCs, motor neurons and fibroblasts to proteasome inhibition using MG132 [39]. As expected, MG132 caused a dose-dependent decrease in viability of each cell type following 16 h treatment; however, pluripotent stem cells were extremely sensitive to MG132 treatment compared to motor neurons or fibroblasts. Cell viability of iPSCs was significantly reduced at MG132 concentrations above 10 nM (*p* < 0.0001, *n* = 3, one-way ANOVA Dunnett’s multiple comparisons test F(9,20) = 222) (Figure 4A). For iPSCs the concentration of MG132 causing toxicity to 50% of the cells (TC_50_) was calculated as 26 nM. MG132 treatment was lethal to 100% of iPSC colonies that detached within hours of treatment with MG132. To confirm whether this sensitivity to MG132 was common to pluripotent stem cells more broadly, or a feature of induced pluripotent stem cells specifically, we confirmed these findings in H9 embryonic stem cells. MG132 significantly reduced cell viability at concentrations above 20 nM (*p* < 0.0001, *n* = 3, one-way ANOVA Dunnett’s multiple comparisons test F(9,20) = 452), with a TC_50_ of 199 nM for MG132 in ESCs (Figure 4A). Motor neurons were more resilient to MG132 proteasomal stress than pluripotent stem cells showing significant reductions in viability at concentrations above 100 μM (*p* < 0.001, *n* = 4, one-way ANOVA Dunnett’s multiple comparisons test F(5,18) = 9.29) (Figure 4A). Fibroblasts were the most resilient to MG132 treatment showing a significant reduction in viability only at 200 μM (*p* < 0.0001, *n* = 4, one-way ANOVA Dunnett’s multiple comparisons test F(5,18) = 10.66)) (Figure 4A). Due to the limit of solubility of MG132, it was not possible to test concentrations of MG132 above 200 μM; therefore, precise TC_50_ values for fibroblasts or motor neurons could not be calculated. Together, our data shows that pluripotent stem cells, of either induced or embryonic origin, are highly sensitive to proteasome inhibition via MG132, indicating a strong dependence on the protein degradation arm of the UPS for pluripotent stem cell survival.

To further investigate the importance of the UPS in motor neuron development, we examined the role of the E1 ubiquitin-activating enzyme UBA1 during motor neuron differentiation. UBA1 is a central regulator of proteostasis, and mutations in UBA1 cause the motor neuron disease spinal muscular atrophy [9,38]. The abundance of UBA1 was significantly increased (36-fold) in the ubiquitome of motor neurons (LFQ = 80,026,250 ± 15,141,763) compared to pluripotent stem cells (LFQ = 2,177,975 ± 678,570) (Figure 2F and Appendix A). Neurite outgrowth was monitored for 24 h following treatment with the UBA1-specific inhibitor PYR41 [40,41] during motor neuron differentiation from iPSCs. We observed significantly reduced neurite length in motor neuron precursors treated with 10 µM PYR41 compared to vehicle control (*p* < 0.05, one-way ANOVA Brown-Forsythe test, F(4,10) = 5.524) (Figure 4B). Longer-term inhibition of UBA1 (three doses of 10 µM PYR41) resulted in 100% cell death by day 6 of the differentiation period (not shown). However, no significant difference in neurite outgrowth was observed following long-term (4 weeks), treatment with low dose (1 µM) PYR41 compared to the vehicle control (not shown). Although no overt morphological differences were observed following four weeks of treatment with 1 µM PYR41, there was a small (1.6-fold) but significant decrease in viability of motor neurons (*p* < 0.05, *n* = 4, two-tailed *t*-test *t* = 4.072 df = 4) (Figure 4C). Together, this data highlights the regulatory role of the UPS and UBA1 in motor neuron differentiation and survival.

## 4. Discussion

We have identified an important role for ubiquitin signaling in both pluripotent stem cell maintenance and motor neuron differentiation, with a broad range of biological functions represented by proteins present in the ubiquitomes of pluripotent stem cells and motor neurons. Proteins involved in DNA replication, ribosomal and spliceosomal proteins were highly enriched in the ubiquitome of pluripotent stem cells, whereas proteins involved in oxidative phosphorylation, extracellular matrix and the cytoskeleton were enriched in the ubiquitome of iPSC-derived motor neurons. Of the proteins identified in motor neurons, >100 play a role in neuronal differentiation. Conversely, in pluripotent stem cells, proteins involved in pluripotency and stem cell proliferation, such as LIN28A, TCEB1, PCNA and SMARCA5, were represented in the ubiquitome. The functional relevance of this analysis is most strikingly demonstrated by the enrichment of UPS components and synaptic proteins in the iPSC-derived motor neuron ubiquitome. Changes in UPS activity have been implicated in regulating differentiation of pluripotent stem cells towards neural lineages. For example, Saez et al. (2018) identified alterations in the levels of E3 ubiquitin ligases during neural differentiation from embryonic stem cells, with these changes modulating pathways including RNA processing and metabolism [42], consistent with our findings.

The ubiquitylation of 18 spliceosomal proteins was significantly downregulated in motor neurons compared to iPSCs. Consistent with this, four of these proteins, TRA2A, TRA2B, U2AF2 and DDX23, have been previously reported as being upregulated in iPSCs [43]. Spliceosomal proteins are modulators of cellular identity that regulate pluripotency and differentiation [44]. Mutant ubiquitin can inhibit splicing events [45] and increasing evidence suggests spliceosomal integrity and maintenance is compromised in motor neuron diseases, including ALS and SMA [46]. Together, these findings present a strong argument for further investigation of UPS regulation of spliceosomal events in motor neurons and motor neuron disease.

Comparison of the ubiquitomes of iPSC with the corresponding differentiated motor neurons revealed a shift in the role of ubiquitin associated with cellular differentiation. Cytoskeletal proteins, such as microtubule-associated proteins and tubulins were detected at significantly higher abundances in the ubiquitome of motor neurons, consistent with previous findings that ubiquitin is an important modulator of cytoskeletal structures [47]. The regulation of synaptic proteins is highly relevant to motor neurons as even small alterations to the ubiquitylation of synaptic proteins can lead to motor neuron death [48]. Ubiquitin signaling regulates synapse formation, and synaptic plasticity (reviewed in [49]). Deregulation of these processes thus leads to neuronal development and neurodegenerative disorders, including autism, motor neuron disease, and brain cancers, such as astrocytoma and medulloblastoma [50]. However ubiquitin signaling also regulates synapse function in a proteasomal degradation-independent manner [51] and likely has roles in motor neurons beyond proteasomal degradation. Future work will dissect the precise signaling pathways of ubiquitylation, either as an activity modifier or a degradation signal, on motor neuron function.

The apparent importance of ubiquitin signaling indicated by differences in the ubiquitomes of iPSCs and motor neurons is also reflected at the functional level, where we observed differences in susceptibility to proteasome inhibition in iPSCs and ESCs compared to motor neurons. Pluripotent stem cells were exquisitely sensitive to proteasome inhibition by MG132, with a TC_50_ in the nanomolar range. UPS inhibition has previously been reported to have a cytotoxic effect in mouse embryonic stem cells, with doses of 200 nM inducing cytotoxic effects [52]. Together these data highlight the importance of the UPS in pluripotent stem cell survival. Thus, an important context for our observations is that the functional outcomes of post-translational modification by ubiquitin extend beyond protein degradation by the proteasome. Further, the ubiquitome detected by our analytical method contains not only ubiquitin-modified proteins, but also ubiquitin-binding proteins that constitute components of the ubiquitylation machinery, and proteins that bind to ubiquitinated proteins [53].

Previous reports have suggested that proteasomal activity is increased in pluripotent stem cells compared to fibroblasts [53], and that this increased activity may be due to an increase in total proteasomal subunits. Vilchez et al. identified a lower level of polyubiquitin chains in embryonic stem cells and an increase in proteasomal activity when normalized to total protein compared to other cell types [54]. An increase in polyubiquitylation is not necessarily indicative of altered proteasomal function, and different poly-Ub chain topologies may signal a range of cellular signaling outcomes [55]. The ratio of proteasomes to total protein is an interesting consideration when comparing different cell types, since total protein changes within cell types [56]. Compared to motor neurons, we observed marginally increased abundance in the iPSCs ubiquitome of the PSMC family that comprise the hexameric ATPases in the catalytic core of the 26S proteasomal subunit. Conversely the motor neuron ubiquitome had significantly increased abundance of multiple members of the PSMD family that comprise the non-ATPase regulatory subunits (or “lid”) of the 26S proteasome, and proteins involved in the deubiquitylating mechanism [57]. High levels of PSMD11 in embryonic stem cells were previously proposed to be responsible for the high proteasome activity in these cells [54]. However, in our dataset, PSMD11 levels were higher in the ubiquitome of motor neurons than iPSCs. The higher abundance of 26S proteasome regulatory subunits in the motor neuron ubiquitome may indicate a high requirement or capacity for degradation of ubiquitylated proteins to maintain protein homeostasis by removing misfolded or damaged proteins in these cells. Of the ubiquitylated proteins identified in human iPSCs in our study, 306 (71%) were also found to be ubiquitylated in the mouse iPSC ubiquitome reported in [25].

In a disease context, multiple lines of evidence suggest a role for ubiquitin signaling changes in ALS, including the accumulation of ubiquitylated protein inclusions within motor neurons. Microarray analysis previously indicated that spinal motor neurons exhibited a downregulation of genes relating to the UPS, as well as mitochondria and RNA metabolism, in spinal cord motor neurons that are sensitive to neurodegeneration in ALS, compared to the more neurodegeneration-resilient ocular motor neurons [22]. This downregulation of UPS genes could be responsible for the greater susceptibility to inhibition of UPS function in motor neuron diseases. We compared the proteins present in the ubiquitome to those known to be associated with ALS; fourteen proteins associated with ALS were present as ubiquitylated in the iPSCs and/or motor neurons. Five of the ALS associated proteins changed significantly between iPSCs and motor neurons; three of these ubiquitylated proteins (dynactin 1, cytoplasmic dynein 1 heavy chain 1, and sequestosome 1) were upregulated in motor neurons, and two (profilin 1 and valosin-containing protein) were upregulated in iPSCs compared to the motor neurons. There have been several UPS components implicated in neurodegenerative diseases, including the deubiquinating components USP14 and cAMP (reviewed in [58]). Strategies using modified antibody fragments have also been used to promote clearance of aggregated TDP43 from the cytosol via the UPS and autophagy–lysosome pathways [59].

Changes in motor neuron excitability in ALS occur prior to the onset of symptoms [60]. These changes are caused by alterations in ion channel expression and function [61] that could be driven by changes to the UPS. We recently showed that the spinal motor neuron proteome is rich in proteins with a high propensity for aggregation and these proteins are downregulated in human patient ALS spinal cord motor neurons [62,63]. In cell models ALS-related mutations cause a depletion of the free ubiquitin pool and increased ubiquitination of proteins involved in oxidative phosphorylation and metabolism [17]. Our work in an ALS mouse model suggests that changes in the ion channels underlying hyperexcitability specifically identify the neurons that are destined to die [61]. Furthermore, our data here confirm a role for the apex E1 ubiquitin-binding enzyme, UBA1, in motor neuron survival and function. Neurite outgrowth following inhibition of UBA1 at 10 µM PYR41 led to neurite outgrowth inhibition and cell death within 6 days, whereas 1 µM did not affect neurite outgrowth but did reduce viability after 4 weeks. Future work may identify a mechanism for the requirement of UBA1 in spinal motor neurons, compared to other more neurodegeneration-resilient cell types. Together, these data suggest that ubiquitin signaling regulates motor neuron function and neurodegeneration and is consistent with the hypothesis that a reduced UPS capacity could drive ALS pathology.

Our data identifies a central and changing role of the ubiquitome during motor neuron differentiation. Given the involvement of UBA1 in this process and its role in motor neuron diseases, understanding the pathways that are most affected under conditions of cellular stress could provide valuable insight into potential therapies. This study suggests that ubiquitin signaling is important in pluripotent stem cell maintenance and motor neuron differentiation and function. Future work should focus on identifying changes to the ubiquitome in response to specific disease mutations or genetic risk factors. The requirement for a functioning ubiquitin proteasome system in motor neurons and their neurites raises the possibility of protecting motor neurons via increasing ubiquitin or ubiquitin-modifying enzymes.

## 5. Conclusions

This study provides the first reported ubiquitomes from human iPSCs compared to differentiated iPSC-derived motor neurons. The relative quantification of these ubiquitylated proteomes revealed significant shifts in the requirements of the UPS contingent with cellular stage. The central UPS regulator, UBA1, spliceosomal and cytoskeletal proteins increased in abundance in the motor neuron ubiquitome compared to pluripotent stem cells. Accordingly, inhibiting UBA1 with low concentrations of PYR41 in motor neurons led to a small but significant reduction in neurite outgrowth and motor neuron viability. Motor neurons were more sensitive to UPS stress compared to fibroblasts. However, we also identified an extreme sensitivity of pluripotent stem cells to UPS stress, indicative of a strong dependence on the UPS for pluripotent stem cell survival.

## Figures and Tables

**Figure 1 cells-08-00581-f001:**
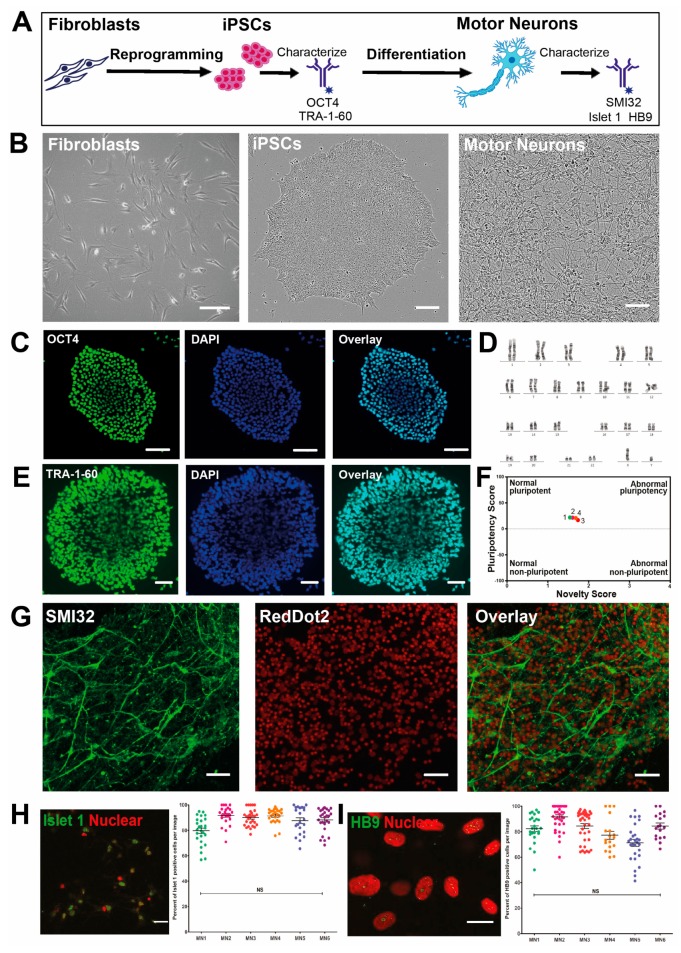
Generation of patient induced pluripotent stem cell (iPSC)-derived motor neurons from fibroblasts. (**A**) Schematic timeline of iPSC reprogramming and motor neuron differentiation. Cells were characterised by immunocytochemistry and ubiquitomics at the pluripotent stem cell stage and motor neuron stage. (**B**) Representative brightfield images of fibroblasts at day 0 of reprogramming, iPSCs on day 2 after passaging, and motor neurons on day 76 of differentiation. Scale bar represents 100 µm. (**C**) Representative epifluorescent images of OCT4 expression (immunofluorescence) in an iPSC colony grown on Matrigel-coated tissue culture plates in TeSR E8, 2 days following passage, with DAPI staining (nucleus). Scale bar represents 100 µm. (**D**) Representative iPSC karyogram confirms the absence of introduced chromosomal abnormalities. (**E**) Representative epifluorescent images of TRA-1-60 expression (immunofluorescence) in an iPSC colony grown on Matrigel-coated tissue culture plates in TeSR E8, 2 days following passage. Scale bar represents 50 µm. (**F**) PluriTest results following comparison of the reprogrammed stem cells to previous characterised cells. (**G**) Representative scanning confocal micrographs of iPSC-derived motor neurons stained showing expression of neuronal marker neurofilament heavy (SMI32) by immunofluorescence, with RedDot2 staining (nucleus). Motor neurons were cultured for 76 days on laminin, collagen I and fibronectin-coated plates. Scale bar represents 100 µm. (**H**) Representative scanning confocal micrographs of iPSC-derived motor neurons stained showing expression of neuronal marker Islet 1 by immunofluorescence, with RedDot2 staining (nucleus), and image quantification via Image J analysis in six cell lines. NS = no significant difference by one-way ANOVA. Scale bar represents 50 µm. (**I**) Representative scanning confocal micrographs of iPSC-derived motor neurons stained showing expression of neuronal marker HB9 by immunofluorescence, with RedDot2 staining (nucleus), and image quantification via Image J analysis in six cell lines. Scale bar represents 25 µm.

**Figure 2 cells-08-00581-f002:**
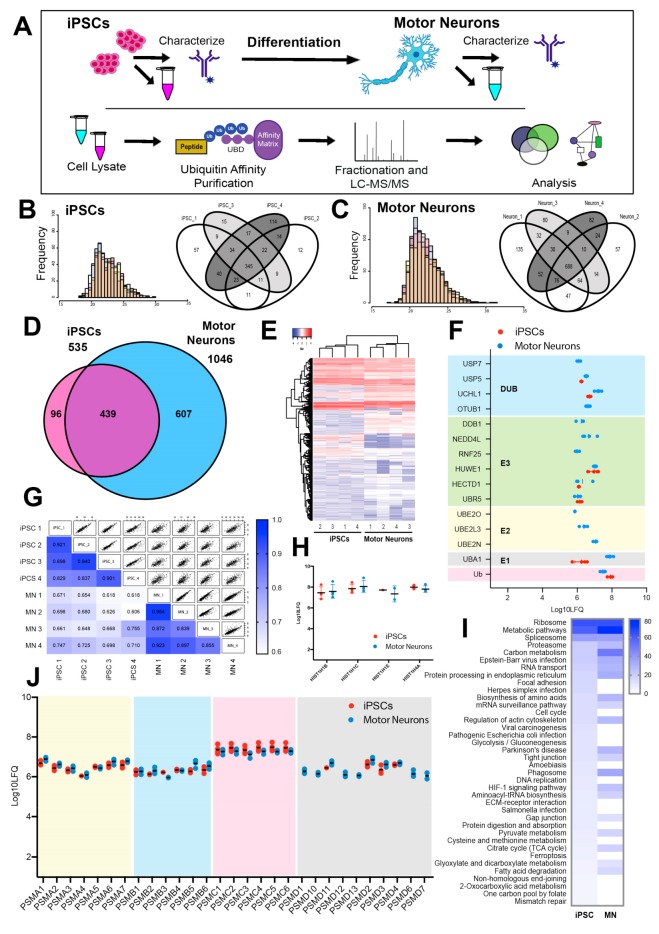
Defining the ubiquitome in iPSCs and iPSC-derived motor neurons. (**A**) Schematic of sample collection from iPSC and iPSC-derived motor neurons (*n* = 4), and processing for analysis. (**B**) Histograms showing distribution of peptide intensities in individual replicates and Venn diagrams showing number of proteins identified in the ubiquitome in individual replicates of iPSC. (**C**) Histograms showing distribution of peptide intensities in individual replicates and Venn diagrams showing number of proteins identified in the ubiquitome in individual replicates of motor neurons. (**D**) Venn diagram showing number of individual proteins identified in the ubiquitome of iPSC and iPSC-derived motor neurons. (**E**) Hierarchical clustering showing relationship of individual replicates of iPSC and motor neuron ubiquitomes. (**F**) Abundance of individual Ub pathway components in the ubiquitome of iPSC and iPSC-derived motor neurons. Data shown are mean ± SEM; *n* = 4. (**G**) Multiple regression analysis of protein abundances (LFQ) in individual replicates, showing Pearson’s correlation values. (**H**) Abundance (LFQ) of histone variants in the ubiquitome of iPSC and motor neurons. (**I**) Heat map showing functional enrichment (using KEGG pathways) in the ubiquitome of iPSC and motor neurons (shading relative to number of individual proteins represented in each pathway). (**J**) Abundance (LFQ) of proteasome subunits in the ubiquitomes of iPSCs and motor neurons.

**Figure 3 cells-08-00581-f003:**
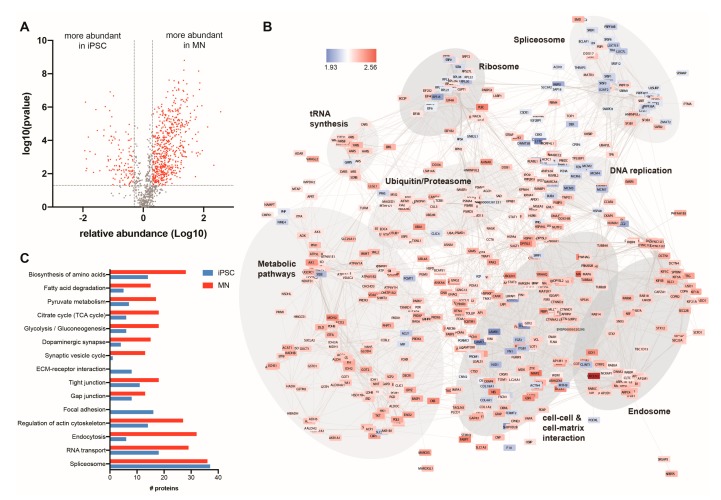
Ubiquitome changes following iPSC differentiation to motor neurons. (**A**) Volcano plot (Log10 abundance vs. −log10 *p*-value) to determine significantly enriched proteins in the ubiquitome of iPSC and iPSC-derived motor neurons. Proteins with adjusted *p* < 0.01 and fold change >2 are shown in red (**B**) Protein–protein interaction map of differentially enriched proteins in the ubiquitome of iPSCs (blue) and motor neurons (red) as determined by STRING analysis (confidence score > 0.700). (**C**) Functional enrichment of differentially expressed proteins within the ubiquitome of iPSCs and motor neurons, showing the number of individual proteins present in each KEGG pathway in each cell state (data from *n* = 4).

**Figure 4 cells-08-00581-f004:**
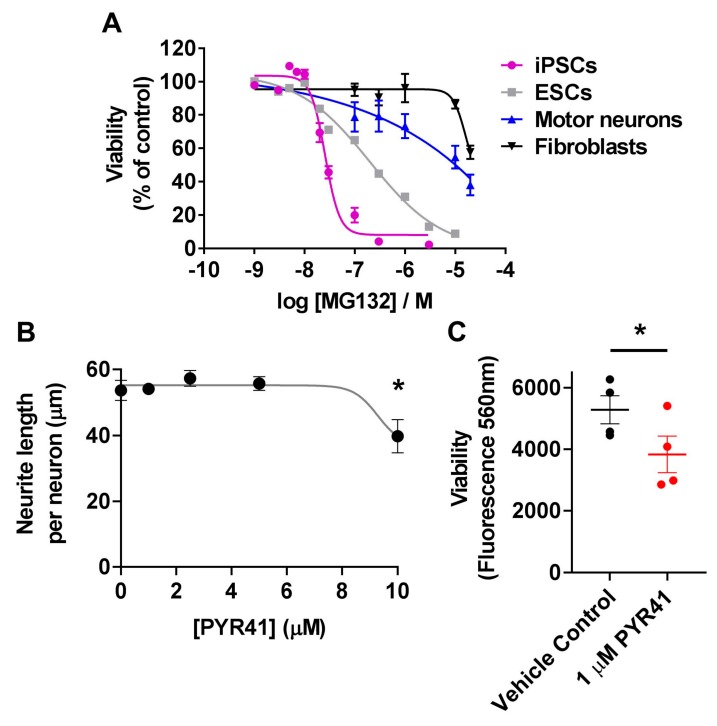
Differential sensitivity of various cell states to UPS inhibition. (**A**) Viability of iPSCs, ESCs, motor neurons and fibroblasts following 16 h treatment with increasing doses of the proteasome inhibitor MG132. (**B**) Neurite outgrowth of motor neuron precursors following 24 h treatment with 1 µM–10 µM UBA1 inhibitor (PYR41). (**C**) Viability of motor neurons subjected to long-term (4 week) treatment with 1 µM PYR41 UBA1 inhibitor (red) or vehicle control (black). Data shown are mean ± SEM, *n* = 3–4, * *p* < 0.05 (one-way ANOVA Brown-Forsythe test (**B**), or two-tailed paired t-test (**C**)).

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
