# Peer review of "The Ubiquitin Proteasome System Is a Key Regulator of Pluripotent Stem Cell Survival and Motor Neuron Differentiation"

_cells, 2019, doi:10.3390/cells8060581_

Round 1
Reviewer 1 Report
The results of the research article entitled “The Ubiquitin Proteasome System is a key regulator of pluripotent stem cell survival and motor neuron differentiation” have been described thoroughly but could be improved with the revision.
1. The importance of previous proteomic studies related to iPSCs pluripotency and their differentiation has not been explored in the introduction part. For example, if available, the authors may include previous pluripotency- and differentiation-related proteomic studies of different types of iPS cells and their differentiation.
2. Section 2.4, Ubiquitomics (in Methods): the authors stated “Mono- and poly-ubiquitylated proteins were identified using ubiquitin affinity matrix and mass-spectrometry as previously described [8,27]”. Although two references have been provided, it would be convenient for the readers to follow this method if the authors describe it briefly in section 2.4.
3. The details of enzyme(s) used in digestion and MaxQuant parameters used in data processing were missing in Methods (Section 2.4)
4. The version or release date (or the number proteins) of Unirpot human database is necessary (Section 2.4)
5. Figure 1A: it has been mentioned a word ‘Characterize’ in Figure 1a twice – need to explain this in figure legend – for example, what kind of characterization?.
6. Figure 2A: this figure shows ‘fractionation and LC-MS/MS’ – it is necessary to provide the details of fractionation in Methods
7. Lines 208-210: what is the rationale here for exploring the histone ubiquitination? The explanation is required for the general readers who are not working in this field.
8. Lines 212-213: what could be the reason for identifying almost 2 times higher number of ubiquitinated proteins in motor neurons when compared with iPSCs? If it is general activation (as mentioned in lines 222-223) then how many of the 439 ubiquitinated proteins (identified in both) were upregulated?
9. Lines 245-248: Figure 2C indicates 439 proteins only were identified in both iPSCs and motor neurons. However, it has been mentioned 585 up-regulated and 120 downregulated between iPSCs and motor neurons according to Fig 3A. Identification number (439) is lower than the quantified proteins (585 + 120). – could you explain it?
10. Lines 300-301: the authors observed 36-fold increase in UBA1 in the ubiquitinome. However, in the abstract (lines 35-36), the authors stated UBA1 was 36-fold up-regulated. Is it UBA1 protein upregulation or the upregulation of ubituitination of UBA1? Needs consistency between the statements. Is there any way to confirm (validation) this upregulation of ubiquitination of UBA1 by performing additional experiment?
11. Have you identified any known ubiquitinated proteins (based on published literature) that were up- or down-regulated between iPSCs and motor neurons?
12. How many of the identified pathways were related to iPSC pluripotency maintenance or associated with differentiation? – it is important to include in discussion part
13. It is important to submit the raw files to public repository (e.g. PRIDE) – only if this journal guide lines asked you to do so.
Author Response
Thank you very much for the positive response to our manuscript. We have addressed all of the reviewer comments and we believe this has improved our manuscript. A point-by-point response to the reviewer requests is included below.
Reviewer 1
The results of the research article entitled “The Ubiquitin Proteasome System is a key regulator of pluripotent stem cell survival and motor neuron differentiation” have been described thoroughly but could be improved with the revision.
1. The importance of previous proteomic studies related to iPSCs pluripotency and their differentiation has not been explored in the introduction part. For example, if available, the authors may include previous pluripotency- and differentiation-related proteomic studies of different types of iPS cells and their differentiation.
The introduction has been expanded to include this topic: ‘Quantitative proteomic analyses provide an unbiased approach allowing the identification of changes during disease or development. In stem cell pluripotency and differentiation, proteomic analyses have highlighted systems-level mechanisms controlling stem cell fate and the coordination of events that are necessary to drive specific development [Lu, 2009]. Using mass spectrometry protein-protein interactions have been identified by combining affinity purification with proteomic identification of interacting proteins. These approaches have increased our understanding of the protein networks involved in pluripotency and cellular differentiation [Pardo, 2010; Wang, 2010].’
2. Section 2.4, Ubiquitomics (in Methods): the authors stated “Mono- and poly-ubiquitylated proteins were identified using ubiquitin affinity matrix and mass-spectrometry as previously described [8,27]”. Although two references have been provided, it would be convenient for the readers to follow this method if the authors describe it briefly in section 2.4.
Methods section now expanded to include a more detailed description of Ubiquitomics technique.
3. The details of enzyme(s) used in digestion and MaxQuant parameters used in data processing were missing in Methods (Section 2.4)
Now included, as above (2).
4. The version or release date (or the number proteins) of Unirpot human database is necessary (Section 2.4)
Now included, as above (2).
5. Figure 1A: it has been mentioned a word ‘Characterize’ in Figure 1a twice – need to explain this in figure legend – for example, what kind of characterization?.
The Figure 1A legend has been amended as suggested: ‘Cells were characterized by immunocytochemistry and ubiquitomics at the pluripotent stem cell stage and motor neuron stage.’
6. Figure 2A: this figure shows ‘fractionation and LC-MS/MS’ – it is necessary to provide the details of fractionation in Methods
Now included, as above (2).
7. Lines 208-210: what is the rationale here for exploring the histone ubiquitination? The explanation is required for the general readers who are not working in this field.
This explanation has been added to the text: Proteomics data is normalised to protein content. This normalisation is evident in the equal representation of various histones in the ubiquitome of iPSC and MN. Ubiquitylated histones represent a large proportion of the total cellular Ubiquitin pool.
8. Lines 212-213: what could be the reason for identifying almost 2 times higher number of ubiquitinated proteins in motor neurons when compared with iPSCs? If it is general activation (as mentioned in lines 222-223) then how many of the 439 ubiquitinated proteins (identified in both) were upregulated?
Refer to quantitative analysis (Fig 3a), and below. The analysis used in figure 2 is non-quantitative, it simply defines a protein as “present” in either condition if detected in a least 2 replicates. No comparison is made of abundance in this analysis.
9. Lines 245-248: Figure 2C indicates 439 proteins only were identified in both iPSCs and motor neurons. However, it has been mentioned 585 up-regulated and 120 downregulated between iPSCs and motor neurons according to Fig 3A. Identification number (439) is lower than the quantified proteins (585 + 120). – could you explain it?
Because the quantitative analysis in Fig3 is a completely different analysis technique. It allows us to identify proteins that are present in both conditions, but with altered abundance. Analysis in Fig 2 defines a protein as “present” in either condition if detected in a least 2 replicates. In contrast, the analysis in Fig 3 uses imputation to include “zero” values (which can't be log transformed) and hence incorporate a broader representation of proteins that would otherwise be excluded in non-quantitative analysis (Fig 2).
10. Lines 300-301: the authors observed 36-fold increase in UBA1 in the ubiquitinome. However, in the abstract (lines 35-36), the authors stated UBA1 was 36-fold up-regulated. Is it UBA1 protein upregulation or the upregulation of ubituitination of UBA1? Needs consistency between the statements. Is there any way to confirm (validation) this upregulation of ubiquitination of UBA1 by performing additional experiment?
Apologies for the confusion, the data refer to increased abundance in the ubqiuitome. This has been clarified in the text to ensure consistency of message between these statements. In the Results: ‘The abundance of UBA1 in the ubiquitome was significantly increased (36 fold) in motor neurons (LFQ = 80026250 +/- 15141763) compared to pluripotent stem cells (LFQ = 2177975 +/- 678570) (Figure 2b and Table S1)’ and in the Abstract (line 36): ‘The UPS regulator, ubiquitin-like modifier activating enzyme 1 (UBA1), was increased 36-fold in the ubiquitome of motor neurons compared to pluripotent stem cells.’
11. Have you identified any known ubiquitinated proteins (based on published literature) that were up- or down-regulated between iPSCs and motor neurons?
Text added to Discussion.
12. How many of the identified pathways were related to iPSC pluripotency maintenance or associated with differentiation? – it is important to include in discussion part
Text added to Discussion.
13. It is important to submit the raw files to public repository (e.g. PRIDE) – only if this journal guide lines asked you to do so.
We agree with the reviewer and these will be available to the public following acceptance of the manuscript.
Reviewer 2 Report
In this paper, Bax et al., examined the importance of proteasome using iPS cell and iPS cell-derived motor neurons. The results showed that both iPS cells and iPS cell-derived motor neurons need the proteasome system general. However, it did not lead to the identification of some UPS-related protein specific to iPS cell-derived motor neurons. I think that it is suitable for publication with only the correction of the text up to Figure 1-3. However, because Figure 4 has a small amount of data, it can not be judged suitable or not suitable. In order to consider the publication of Figure 4, I recommend the re-examination by additional experiments.
…………………………………………………………
1. Introduction
It is recommended to change the order of sentences.
Also, please add sentences as necessary.
1, what is Ub.
2, What is UPS. Please show some research method of UPS.
3, The types of diseases that related with UPS (It is recommended to create a table).
4, The type of motor nerve disease that related with UPS (It is recommended to create a table).
5, About the method of treatment of motor nerve disease relating UPS.
6, What are ES cells and iPS cells.
7, Progress of research on UPS of ES cells and iPS cells.
8, Presence or absence of UPS disease elucidated by iPS cell pathological model.
9, Presence or absence of drug related with UPS disease developed in iPS cell drug discovery.
10, About the perfection degree of recent iPS cell origin motor nerve.
11, The purpose of this study using iPS cell-derived motor nerve.
…………………………………………………………
1. Materials and Methods (2.1. Generation of iPSC-derived motor neurons)
The title should be written as “2.1. Generation of iPSC”.
…………………………………………………………
Line 102-104
“Fibroblasts were reprogrammed into induced pluripotent stem cells using microRNA enhanced mRNA reprogramming as per manufacturer’s recommendations (Stemgent, Cambridge, Massachusetts, United States) and characterise as per [23,24].”
In reference 23 the authors used the following reprogramming reagents.
”StemMACS mRNA Reprogrammingkit (Miltenyi Biotec, #130-104-460)””University of WollongongHuman Research Ethics Committee (HE13/299)”
In reference 24 the authors used the following reprogramming reagents.
”StemgentmicroRNA-Enhanced mRNA Reprogramming Kit (Stemgent, 00–0071)””University of Wollongong Human Ethics Committee (HE 13/299)”
Analysis data of iPS cells are greatly influenced by donor age and initialization method. Therefore, the authors should provide detailed information in this paper. In addition, the original article (or patent number) and design of the mRNA reprogramming vector should be shown.
…………………………………………………………
1. Materials and Methods
2.2. Motor neuron differentiation and characterisation
Please show all reagent names used for differentiation induction. Also, add the product number as much as possible.
Please illustrate the three typical differentiation induction methods for iPS cell-derived motor neurons that have been reported so far. Next, please show the method of differentiation induction used by the authors as detailed as possible in Figure.
…………………………………………………………
3. Results
3.1. High yield differentiation of motor neurons from fibroblast-derived iPSCs.
Line 172-174
“To investigate molecular mechanisms underlying differentiation of motor neurons in development, and as potential models for diseases involving motor neurons (e.g. ALS), fibroblasts from six donors were reprogrammed to iPSCs (Figure 1A) [20][18].”
This sentence should be changing, because the reader will make misunderstands that iPS cells made from ALS patients.
Please add the health status, age, and gender of the six donors to “2.1. Generation of iPSC”.
…………………………………………………………
Figure 1A: Please add the target names (OCT4, HB9, SMI32) to the drawing of the antibody.
Please add the explanatory note of (E) to the figure legend.
Please delete Figure 1D. And add the all data of Supplementary Figure 1 into Figure 1.
…………………………………………………………
3. Results
3.2. The ubiquitin-modified proteome (ubiquitome) of iPSCs and motor neurons
iPS cells and neurons differ in cell size and morphology. Therefore, it is necessary to correct the data by the amount of expression such as housekeeping gene. Please add how you corrected the data in text. Supplementary Figure 2's proofreading are bad (the letters of E and F are large, the letter of G is small) (Supplementary figure 2C and 2G is two)
…………………………………………………………
Line 203-204
“We also observed the significant overlap in proteins identified in individual replicates (Supplementary Figure 2A-B)”
Authors should be cautious in handling this data. The authors describe the iPSC expressed protein noumbers as "535" and the motor nerve expressed protein noumbers as "1045" (Figure 2C). However, the numbers of iPSC expression proteins detected in quadruple are 345, and that of motor neurons is 688 (Supplementary Figure 2A-B). Authors should add the process by which the numbers 535 and 1045 are calculated.
…………………………………………………………
Please add the all data of Supplementary Figure 2 into Figure 2.
…………………………………………………………
Line 221-223
“The presence of significantly elevated levels of Ub-associated forms of a number of key enzymes at each level in the Ub conjugation hierarchy (E1, E2, E3 and DUB) in motor neurons (Figure 2D) suggests a general activation of the Ub system in the differentiated cell state.”
Please move the above sentence before the next one.
”Importantly, we observed very little difference (Line 215) ”
…………………………………………………………
Please replace Supplementary Figure 3 as the image resolution does not meet publication quality.
…………………………………………………………
3. Results
3.4. Differential sensitivity of iPSC and motor neurons to UPS inhibition.
DMSO has a potent cell death-inducing effect on iPS cells.
Controls should be compared at the same (or higher) dose of DMSO.
Describe the DMSO dosage (µl/ml) for each reagent concentration.
…………………………………………………………
4. Discussion
There are too many cells general content about UPS. The discussion should be limited to undifferentiated cells, nerve cells and nerve lesions.
Figure 2C: If there are some factors involved in UPS that is expressed only in motor nerves, please add a comment.
Please add the UPS related protein's information that involved in motor nerve disease.
Please add comments on TDP-43 degraded by the proteasome on the pathogenesis of ALS.
(Scientific Reportsvolume 8, Article number: 6030 (2018))
…………………………………………………………
5. Conclusions
The drug effect of PYR41 is considered to be rather weak (Figure 4C: 10µM x 24h, Figure 4D: 1µM x 4 week).
Therefore, I think that the author's conclusion is over-discussion.
At present, we can only draw conclusions about the data shown in Figure 1- 3.
Author Response
Thank you very much for the positive response to our manuscript. We have addressed all of the reviewer comments and we believe this has improved our manuscript. A point-by-point response to the reviewer requests is included below.
Reviewer 2
In this paper, Bax et al., examined the importance of proteasome using iPS cell and iPS cell-derived motor neurons. The results showed that both iPS cells and iPS cell-derived motor neurons need the proteasome system general. However, it did not lead to the identification of some UPS-related protein specific to iPS cell-derived motor neurons. I think that it is suitable for publication with only the correction of the text up to Figure 1-3. However, because Figure 4 has a small amount of data, it can not be judged suitable or not suitable. In order to consider the publication of Figure 4, I recommend the re-examination by additional experiments.
…………………………………………………………
1. Introduction
It is recommended to change the order of sentences.
Amended as requested.
Also, please add sentences as necessary.
1, what is Ub. Now included in introduction.
2, What is UPS. Please show some research method of UPS. Included.
3, The types of diseases that related with UPS Now included in introduction.
4, The type of motor nerve disease that related with UPS (It is recommended to create a table). Now included in introduction.
5, About the method of treatment of motor nerve disease relating UPS.
There have been several targets identified in the UPS to treat neurodegenerative diseases, including the DUB USP14 and cAMP (reviewed in Myeku & Duff 2018 Trends Mol Med) – included in Discussion
6, What are ES cells and iPS cells. Now included in introduction.
7, Progress of research on UPS of ES cells and iPS cells. Now included in introduction.
8, Presence or absence of UPS disease elucidated by iPS cell pathological model.
9, Presence or absence of drug related with UPS disease developed in iPS cell drug discovery.
For points 8 and 9, there are no studies that we are aware of, hence why we believe our manuscript provides some interesting insight.
10, About the perfection degree of recent iPS cell origin motor nerve. Now included.
11, The purpose of this study using iPS cell-derived motor nerve. Now included.
…………………………………………………………
1. Materials and Methods (2.1. Generation of iPSC-derived motor neurons)
The title should be written as “2.1. Generation of iPSC”.
…………………………………………………………
Line 102-104
“Fibroblasts were reprogrammed into induced pluripotent stem cells using microRNA enhanced mRNA reprogramming as per manufacturer’s recommendations (Stemgent, Cambridge, Massachusetts, United States) and characterise as per [23,24].”
In reference 23 the authors used the following reprogramming reagents.
”StemMACS mRNA Reprogrammingkit (Miltenyi Biotec, #130-104-460)””University of WollongongHuman Research Ethics Committee (HE13/299)”
In reference 24 the authors used the following reprogramming reagents.
”StemgentmicroRNA-Enhanced mRNA Reprogramming Kit (Stemgent, 00–0071)””University of Wollongong Human Ethics Committee (HE 13/299)”
Analysis data of iPS cells are greatly influenced by donor age and initialization method. Therefore, the authors should provide detailed information in this paper. In addition, the original article (or patent number) and design of the mRNA reprogramming vector should be shown.
This section has been expanded with the required information.
…………………………………………………………
1. Materials and Methods
2.2. Motor neuron differentiation and characterisation
Please show all reagent names used for differentiation induction. Also, add the product number as much as possible.
Included as appropriate. Please note the protocol used is well-established in the literature and the publications are referenced (line 143).
Please illustrate the three typical differentiation induction methods for iPS cell-derived motor neurons that have been reported so far. Next, please show the method of differentiation induction used by the authors as detailed as possible in Figure.
More details included in the Introduction.
…………………………………………………………
3. Results
3.1. High yield differentiation of motor neurons from fibroblast-derived iPSCs.
Line 172-174
“To investigate molecular mechanisms underlying differentiation of motor neurons in development, and as potential models for diseases involving motor neurons (e.g. ALS), fibroblasts from six donors were reprogrammed to iPSCs (Figure 1A) [20][18].”
This sentence should be changing, because the reader will make misunderstands that iPS cells made from ALS patients.
Text amended.
Please add the health status, age, and gender of the six donors to “2.1. Generation of iPSC”.
Amended as requested.
…………………………………………………………
Figure 1A: Please add the target names (OCT4, HB9, SMI32) to the drawing of the antibody.
Please add the explanatory note of (E) to the figure legend.
Please delete Figure 1D. And add the all data of Supplementary Figure 1 into Figure 1.
Amended as requested.
…………………………………………………………
3. Results
3.2. The ubiquitin-modified proteome (ubiquitome) of iPSCs and motor neurons
iPS cells and neurons differ in cell size and morphology. Therefore, it is necessary to correct the data by the amount of expression such as housekeeping gene. Please add how you corrected the data in text.
Now included, as above (2). “Housekeeping” genes are not useful in this context as there are such widespread gene expression changes during differentiation. Proteomics data is normalised to protein content. This normalisation is evident in the equal representation of various histones in the ubiquitome of iPSC and MN.
Supplementary Figure 2's proofreading are bad (the letters of E and F are large, the letter of G is small) (Supplementary figure 2C and 2G is two)
Supp Fig 2 has been edited as per the requests of Reviewer 1 and is now part of Fig. 2. We confirm the letters within Figures are the same size.
…………………………………………………………
Line 203-204
“We also observed the significant overlap in proteins identified in individual replicates (Supplementary Figure 2A-B)”
Authors should be cautious in handling this data. The authors describe the iPSC expressed protein noumbers as "535" and the motor nerve expressed protein noumbers as "1045" (Figure 2C). However, the numbers of iPSC expression proteins detected in quadruple are 345, and that of motor neurons is 688 (Supplementary Figure 2A-B). Authors should add the process by which the numbers 535 and 1045 are calculated.
See (8) and (9) above. For non-quantitative analysis, proteins were defined as “present” in either condition if detected in a least 2 replicates. This description has now been added to the methods section and relevant figure caption.
…………………………………………………………
Please add the all data of Supplementary Figure 2 into Figure 2.
Amended as requested.
…………………………………………………………
Line 221-223
“The presence of significantly elevated levels of Ub-associated forms of a number of key enzymes at each level in the Ub conjugation hierarchy (E1, E2, E3 and DUB) in motor neurons (Figure 2D) suggests a general activation of the Ub system in the differentiated cell state.”
Please move the above sentence before the next one.
”Importantly, we observed very little difference (Line 215) ”
Amended as requested.
…………………………………………………………
Please replace Supplementary Figure 3 as the image resolution does not meet publication quality.
We can only assume that the upload/download affected the quality of the image, we confirm the image resolution of the final image meets publication quality.
…………………………………………………………
3. Results
3.4. Differential sensitivity of iPSC and motor neurons to UPS inhibition.
DMSO has a potent cell death-inducing effect on iPS cells.
Controls should be compared at the same (or higher) dose of DMSO.
Describe the DMSO dosage (µl/ml) for each reagent concentration.
We confirm that the DMSO vehicle control was used at the same concentration as the drugs and was kept at 0.1%. This has been added to the text. Methods Section 2.5: ‘In the vehicle control DMSO was used at the same final concentration in media (0.1%) as in the MG132 treatments’ and ‘In the vehicle control DMSO was used at the same final concentration in media (0.1%) as in the PYR41 treatments’.
…………………………………………………………
4. Discussion
There are too many cells general content about UPS. The discussion should be limited to undifferentiated cells, nerve cells and nerve lesions.
Discussion focuses on pluripotent stem cells, motor neurons and disease.
Figure 2C: If there are some factors involved in UPS that is expressed only in motor nerves, please add a comment.
We are not aware of UPS components that are specific to motor neurons.
Please add the UPS related protein's information that involved in motor nerve disease.
This is included.
Please add comments on TDP-43 degraded by the proteasome on the pathogenesis of ALS.
(Scientific Reportsvolume 8, Article number: 6030 (2018))
Added.
…………………………………………………………
5. Conclusions
The drug effect of PYR41 is considered to be rather weak (Figure 4C: 10µM x 24h, Figure 4D: 1µM x 4 week).
Therefore, I think that the author's conclusion is over-discussion.
At present, we can only draw conclusions about the data shown in Figure 1- 3.
The effect of PYR41 on neurons is significantly different to controls at low concentrations of drug (10 µM Fig 4B (previously Fig4C) and 1 µM Fig 4C (previously Fig4D)). This information was previously in the text, however in order to make this abundantly clear we have amended the Figures to show the significance on the Figures and amended the text ‘We observed significantly reduced neurite length in motor neuron precursors treated with 10 µM PYR41 compared to vehicle control (p<0.05, one-way ANOVA Brown-Forsythe test, F(4,10)=5.524)’. Significant reductions in neurite length and viability over the course of the experiments demonstrate an effect on neuron function and survival. As we previously described longer-term inhibition of UBA1 (3x doses of 10 µM PYR41) resulted in all cells dying by day 6 of the differentiation period (hence this is a very significant and unreported effect of PYR41). However, no significant difference in neurite outgrowth was observed following long-term (4 weeks), treatment with low dose (1 µM) PYR41 compared to the vehicle control. The t test details are now included: ‘small (1.6 fold) but significant decrease in viability of motor neurons (p<0.05, n=4, two-tailed t-test t=4.072 df=4)’. We agree with the reviewer that future work should focus on the precise role of UBA1 in motor neuron survival / neurodegeneration. For these reasons we have changed the text in line with these points. Section 5 Conclusions: ‘Accordingly inhibiting UBA1 with low concentrations of PYR41 in motor neurons led to a small but significant reduction in neurite outgrowth and motor neuron viability.’ We have also changed the final sentence of the abstract in line with this request.